# Deep-Learning for Tidemark Segmentation in Human Osteochondral Tissues Imaged with Micro-computed Tomography

Aleksei Tiulpin[1,2], Mikko Finnilä[1], Petri Lehenkari[1,2], Heikki J. Nieminen[1,3,4], and Simo Saarakkala[1,2]

[1] University of Oulu, Finland
[2] Oulu University Hospital, Finland
[3] University of Helsinki, Helsinki, Finland
[4] Aalto University, Espoo, Finland

**Abstract.** Three-dimensional (3D) semi-quantitative grading of pathological features in articular cartilage (AC) offers significant improvements in basic research of osteoarthritis (OA). We have earlier developed the 3D protocol for imaging of AC and its structures which includes staining of the sample with a contrast agent (phosphotungstic acid, PTA) and a consequent scanning with micro-computed tomography. Such a protocol was designed to provide X-ray attenuation contrast to visualize AC structure. However, at the same time, this protocol has one major disadvantage: the loss of contrast at the tidemark (calcified cartilage interface, CCI). An accurate segmentation of CCI can be very important for understanding the etiology of OA and *ex-vivo* evaluation of tidemark condition at early OA stages. In this paper, we present the first application of Deep Learning to PTA-stained osteochondral samples that allows to perform tidemark segmentation in a fully-automatic manner. Our method is based on U-Net trained using a combination of binary cross-entropy and soft-Jaccard loss. On cross-validation, this approach yielded intersection over the union of 0.59, 0.70, 0.79, 0.83 and 0.86 within 15 $\mu m$, 30 $\mu m$, 45 $\mu m$, 60 $\mu m$ and 75 $\mu m$ padded zones around the tidemark, respectively. Our codes and the dataset that consisted of 35 PTA-stained human AC samples are made publicly available together with the segmentation masks to facilitate the development of biomedical image segmentation methods.

**Keywords:** Osteoarthritis, 3D Histology, Deep Learning

## 1 Introduction

Osteoarthritis (OA) is a common field of interest in micro-computed tomography ($\mu$CT) research. OA is primarily characterized by progressive degeneration of structure and composition articular cartilage (AC), along with the sclerotic changes in subchondral bone [4]. These changes in the microstructure of AC and subchondral bone can be visualized in three-dimensions (3D) using $\mu$CT.

Conventionally, without any external X-ray contrast agents or sample processing protocols, only calcified tissue can be visualized. Thus, direct $\mu$CT imaging of soft tissues, such us AC, is not possible. To mitigate this limitation of X-ray imaging, several contrast agents have been introduced to provide X-ray attenuation contrast for the AC, such as phosphotungstic acid (PTA), CA4+ and others [14,10,6].

Specifically for OA, a novel *ex-vivo* $\mu$CT contrast method and a protocol to quantify collagen distribution in AC has recently been introduced along with the 3D grading system [10,11]. There, PTA was validated as a contrast agent, since it directly binds to collagen and significantly increases the attenuation contrast within the cartilage tissue [11,7]. However, despite the unique possibility to image soft tissues, PTA staining has one major drawback when it is used for osteochondral tissue: X-ray attenuation contrast at the tidemark (calcified cartilage interface; CCI) is lost due to the accumulation of PTA. Another drawback of the PTA staining is the occasional occurrence of non-enhancing regions, *i.e.* voids, at the CCI [10]. Both of these limitations and the typical examples of the PTA-stained samples analyzed in this study are illustrated in Figure 1.

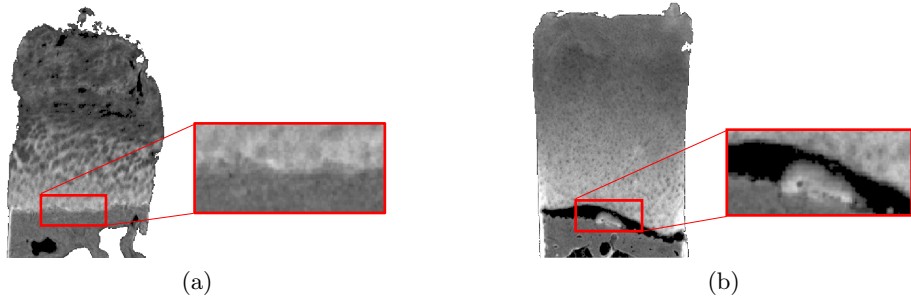

(a)                                     (b)

**Fig. 1.** Examples of the slices from $\varnothing = 2mm$ human osteochondral plugs imaged with contrast-enhanced $\mu$CT. a) a typical sample showing the loss of the contrast at the CCI. b) a typical non-enhancing region (void) which occurs with some samples.

An accurate analysis of CCI from PTA-stained $\mu$CT image stacks is of high importance in the evaluation of early OA-induced changes [8]. Two straightforward solutions exist: either to perform a manual annotation of this area, or, alternatively, perform double imaging – with and without PTA. However, both of these options are time consuming and could be avoided with the help of Machine Learning. In clinical OA research Machine Learning is has been applied to various tasks [21,20,19,12,15,2], however, its application in OA basic research so far has been limited [1].

Recently, one form of Machine Learning – Deep Learning (DL) has become a gold standard in medical image segmentation [17]. Fully-convolutional neural networks (CNN) have shown drastic improvements in the performance of the segmentation methods and decreased their computational time [17]. In particular,

U-Net CNN architecture [16] allowed to significantly improve the bio-medical image segmentation.

In this study, we tackled the problem of automatic tidemark segmentation in PTA-stained osteochondral samples using Deep Learning. This study has the following contributions:

– We present a method based on Deep Learning that allows to perform assessment of tidemark in PTA-stained human osteochondral samples.
– We also present a data acquisition protocol based that allowed to obtain the segmentation masks without their explicit annotation by a human expert.
– In our experiments, we demonstrated the performance of popular U-Net architecture and assessed binary cross-entropy, focal and soft-Jaccard losses.
– Finally, we release our source code and the dataset with the ground truth masks for the benefit of the community.

## 2    Materials and methods

Our imaging pipeline consisted of sample preparation, imaging, data pre-processing and, finally, image segmentation. The graphical illustration of this process is demonstrated in Figure 2 and also in Figure 3, respectively. The following subsections describe our methodology in details.

### 2.1    Samples preparation and imaging protocol

We followed the institutional guidelines and regulations (Institutional ethics approval PPSHP 78/2013, The Northern Ostrobothnia Hospital District's ethical comittee) during sample extraction. The samples were obtained from $n = 20$ patients undergoing total knee arthroplasty surgery (informed consents obtained). At the preparation stage, the osteochondral plugs ($\varnothing = 2mm$, depth $\approx 4$ mm) were drilled from tibial and femoral condyles. These plugs were then frozen under $-80°$C. Before the imaging, we thawed the osteochondral plugs and fixed them in 10% neutral-buffered formalin for a minimum of 5 days. Subsequently, these plugs were wrapped into parafilm and orthodonic wax to avoid sample drying during the imaging process.

At first, we stained the samples with CA4+ contrast agent and imaged them using a $\mu$CT system (Bruker microCT Skyscan 1272, Kontich, Belgium; 45 kV, 222 $\mu$A, 3.2 $\mu$m voxel side length, 3050 ms, 2 frames/projection, 1200 projections, $0.25mm$ aluminum filter) to be used in another study. After the imaging, CA4+ was washed out and the plugs were stained in PTA for 48 hours before the second round of imaging with $\mu$CT using the same imaging settings.

Both CA4+ and PTA data were reconstructed using NRecon software of version 1.6.10.4; Bruker microCT, Kontich, Belgium. Eventually, these 3D stacks were co-registered using rigid intensity-based registration (mean squared error loss) with a subsequent manual adjustment. Subsequently, CA4+ stacks' intensities were thresholded to obtain the hard tissue masks used as segmentation

ground truth. At the final step of the process, we graded each individual cartilage feature from PTA-stained samples according to the 3D histopathological grading system [10].

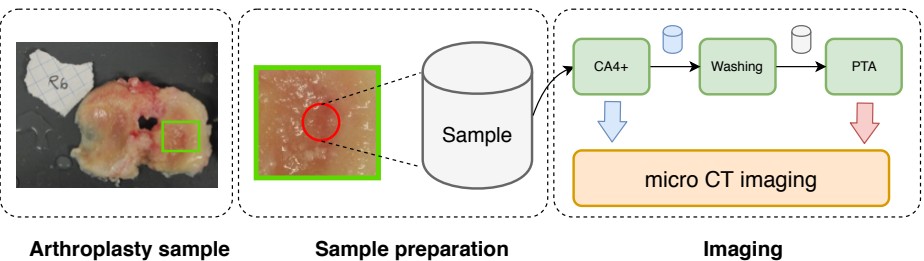

**Fig. 2.** Data acquisition pipeline: from sample preparation to imaging.

## 2.2   Data pre-processing

Our imaging protocol allowed to obtain the 3D volumes of human cartilage and the mask annotations for the underlying mineralized tissues. The original size of the reconstructed samples ranges from $756 \times 756$ to $1008 \times 1008$ pixels in width and 884 to 2067 pixels in height (including the empty space around the sample). To harmonize the data and reduce its size, we firstly cut the bottom 30% of the scanned volume and performed a global contrast normalization of its intensities to $[0, 1]$ range. Subsequently, we performed a thresholding with a cut-off 0.1 and summed all the intensities of the obtained volume along the Z-axis. We used active contours method from OpenCV [3] to identify the largest closed contour in the obtained summed image and then identified its center of mass.

Having the center of mass of the sample in XY plane, we performed the cropping of the original volumes and the corresponding ground truth masks to the size of $448 \times 448 \times 768$ (XYZ) voxels. All the volumes and their masks were then split into ZX and ZY slices to enlarge the dataset in slice-wise segmentation done by a U-Net-like Deep Neural Network [16].

## 2.3   Network Architecture

Our model is inspired by U-Net [16] with minor modifications. Here, we used 24 convolutional filters as the base width of our model and doubled this quantity every time after the max-pooling layer. The depth of the model was set to 6 and bilinear interpolation was used in the decoder of our model. Finally, every convolutional module of the model had two consequent blocks of convolution, batch normalization and ReLU layers.

## 2.4   Loss function

In this study, we evaluated several loss functions. As such, we investigated Binary Cross-Entropy (BCE), soft-Jaccard loss $(1 - J; J$ – soft-Jaccard index), focal loss

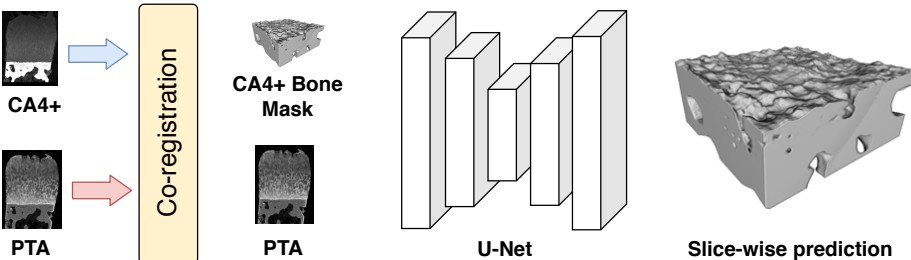

**Fig. 3.** Data processing pipeline. Here, we co-registered CA4+ and PTA samples and obtained the segmentation masks for hard tissues. These masks were used in training of our segmentation model.

and also a combination of BCE and soft jaccard losses. Instead of computing a direct sum of BCE and soft-Jaccard losses, Iglovikov *et al.* [5] proposed to combine BCE and a negative of $\log \mathrm{J}$:

$$L(\mathbf{w}, \mathbf{X}, \mathbf{y}) = \mathrm{BCE}(\mathbf{w}, \mathbf{X}, \mathbf{y}) - \log \mathrm{J}(\mathbf{w}, \mathbf{X}, \mathbf{y}), \tag{1}$$

where $\mathbf{w}$ are the model's weights, $\mathbf{X}$ are the images and $\mathbf{y}$ are the ground truth segmentation masks. We found that the loss in equation 1 yields better performance than when computing soft-Jaccard without a logarithm.

### 2.5   Evaluation metric

As a main evaluation metric, we used Jaccard coefficent (intersection over the union, IoU). IoU was computed only at the area padded around the tidemark. In particular, we identified the location of the tidemark slice-by-slice and for every slice we created a padded region of $\pm P$ pixels. Such masks allowed to estimate the IoU only within the zone of the interest ignoring the other, non-relevant parts of the sample, e.g. bone. Besides the IoU, we also computed the complimentary metrics: Dice's and Volumetric similarity scores.

## 3   Experiments

### 3.1   Implementation details

We implemented our models and training pipelines using PyTorch [13].To augment our data, we applied random cropping, horizontal flip and random gamma-correction, varying value of gamma from 0.5 to 2. To make our model applicable to the real-life scenario when the black edges (air around the sample) are seen in the full sample, we first performed a padding to $800 \times 800$ pixels before random cropping. For the validation set, we used the original size of the images of $768 \times 448$. We used SOLT library [18] to perform data augmentation.

All our experiments were conducted with Adam optimizer [9], batch size of 32, learning rate of $1e-4$ and a weight decay of $1e-4$. For the focal loss, we used the standard hyperparameters: $\alpha = 0.25$ and $\gamma = 2$. All the experiments were done using group-5-fold stratified cross-validation, where the group division was performed by subject id and stratification was done using the previously mentioned 3D histopathological grades obtained for the calcified zone [10].

We assessed the results on sample-wise out-of-fold predictions. Here, we averaged the inference results for each sample's ZX and ZY slices and thresholded the obtained masks with the threshold of 0.3 for the combined loss and 0.5 for BCE and focal losses, respectively. The padding values $P$ for computing the IoU were set to 15 $\mu m$, 30 $\mu m$, 45 $\mu m$, 60 $\mu m$, 75 $\mu m$, 90 $\mu m$, 105 $\mu m$, 120 $\mu m$, 135 $\mu m$ and 150 $\mu m$.

### 3.2 Segmentation performance

The performance of our network with different loss functions on cross-validation for IoU, Dice's and Volumetric similarity scores is presented in Figure 4.

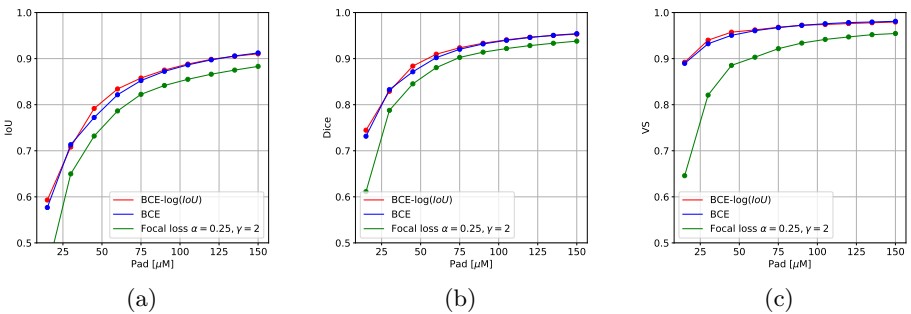

(a)                       (b)                       (c)

**Fig. 4.** Median values of performance metrics for different levels of padding around the tidemark. Here, subplots (a), (b) and (c) show the performance for IoU, Dice and Volumetric similarity scores, respectively.

More fine-grained assessment of the median values of the performance metrics and their standard deviations is presented in Table 1. From Figure 4 and Table 1 it can be seen that for all the metrics, a combination of BCE and jaccard losses from equation 1 yields better performance in the close proximity to the tidemark.

## 4   Conclusion

In this study, we for the first time applied Deep Learning to $\mu CT$ imaged osteo-chondral samples in order to segment the tidemark. The results presented in this paper are promising and indicate the possibility of accurate CCI segmentation even with a 2-dimensional method. Despite this, we believe that the presented

**Table 1.** Median and standard deviation of IoU for different levels of tidemark padding.

| Loss | Pad [$\mu m$] | | | | |
|---|---|---|---|---|---|
| | **15** | **30** | **45** | **60** | **75** |
| BCE | $0.57 \pm 0.14$ | $\mathbf{0.71 \pm 0.11}$ | $0.77 \pm 0.10$ | $0.82 \pm 0.09$ | $0.85 \pm 0.08$ |
| Focal | $0.44 \pm 0.19$ | $0.65 \pm 0.18$ | $0.73 \pm 0.15$ | $0.79 \pm 0.14$ | $0.82 \pm 0.12$ |
| BCE-log(Jaccard) | $\mathbf{0.59 \pm 0.13}$ | $0.70 \pm 0.10$ | $\mathbf{0.79 \pm 0.08}$ | $\mathbf{0.83 \pm 0.08}$ | $\mathbf{0.86 \pm 0.07}$ |

results can further be improved. In particular, we think that a optimizing the segmentation of the tidemark directly with a volumetric model, e.g. 3D U-Net could yield better results. Finally, the future studies should also leverage other, surface-related metrics, e.g. hausdorff distance for more precise assessment of the segmentation results. The codes and the dataset are released on the project's GitHub page: https://github.com/MIPT-Oulu/mCTSegmentation.

## 5    Acknowledgements

This work was supported by Academy of Finland (grants 268378, 303786, 311586 and 314286), European Research Council under the European Union's Seventh Framework Programme (FP/2007-2013)/ERC Grant Agreement no. 336267, the strategic funding of the University of Oulu and KAUTE foundation. We would also like to acknowledge CSC IT Center for Science, Finland, for generous computational resources. Tuomas Frondelius is acknowledged for the initial experiments with the data and Santeri Rytky is acknowledged for the useful comments and proofreading of the paper.

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
