# OpenReview forum: "Deep-Learning for Tidemark Segmentation in Human Osteochondral Tissues Imaged with Micro-computed Tomography"
_MICCAI.org/2019/Workshop/COMPAY — Submitted to COMPAY 2019_

### Official Review · AnonReviewer1 · 2019-08-13
**Concern about the novelty of this work**

**Rating:** 5
**Confidence:** 3

**Review:**

This manuscript shows that an optimized Unet is able to perform tidemark segmentation on PTA-stained osteochondral samples. The authors provided the background in details which is good for the reader to understand the problem. While the problem to be solved is new, the employed methodology is not new. A few concerns regard to the paper are as follows.
1. It would be better to provide the rationale behind eq 1. Why it is better than other losses in this problem?
2. Lack of contribution in terms of the methodology.
3. Should consider increasing the sample size. The sample size in the manuscript is too small, n=34 patient-wise.
4. Section 2.2, why 30% was used?
5. Section 2.2, what is the reason to split the volumes into ZX and ZY slices but not other dimensions?

---

### Official Review · AnonReviewer4 · 2019-08-15

**Rating:** 5
**Confidence:** 3

**Review:**

A large dataset is very beneficial to the community. But the novelty of the method is a bit lacking.
Please explain the global contrast normalization in the data pre-processing step.
Please also explain the motivation of using BCE-log
Looks like an advanced data argumentation tool is used. Can the authors discuss which kind of argumentation is useful?